# PERSONAS AS A WAY TO MODEL TRUTHFULNESS IN LANGUAGE MODELS

## ABSTRACT

Large Language Models (LLMs) are trained on vast amounts of text from the internet, which contains both factual and misleading information about the world. *Can language models discern truth from falsehood in this contradicting data?* Expanding on the view that LLMs can model different communicative agents, we present the *persona hypothesis*: LLMs can cluster agents into *personas* using common features of their generations. For instance, a truthful persona is a group of agents that are likely to produce truthful text and that share similar features like formal writing styles and scientific references. By modeling this persona, LLMs can generalize truthfulness beyond the specific contexts in which each agent generated the training text. For example, the model can infer that the agent "Wikipedia" will behave truthfully on topics that were only generated by "Science" because they both belong to the truthful persona. We show evidence for the persona hypothesis via two observations: (1) we can probe whether a model's answer will be truthful before it is generated; (2) finetuning a model on a set of facts improves its truthfulness on unseen topics. Next, using arithmetics as a synthetic environment, we show that language models can separate true and false statements, and generalize truthfulness across agents; but only if agents in the training data share a truthful generative process that enables the creation of a truthful persona. Overall, our findings suggest that models can exploit hierarchical structures in the data to learn abstract concepts like truthfulness.

## 1 INTRODUCTION

Large Language Models (LLMs) are pretrained on increasing amounts of data from the internet (Brown et al., 2020; Chowdhery et al., 2022)—a noisy, and mostly uncurated corpus—which contains both truthful statements about the world and untruthful statements such as misconceptions and conspiracy theories. The false claims in the data pose a risk of misinformation as they can be propogated by the model (Lin et al., 2021). Intriguingly, recent work shows that the truth value of a statement can be elicited from its embeddings (Burns et al., 2022; Li et al., 2023). This motivates the main question of this work: how do LLMs distinguish truth from falsehood?

Consider two contradicting statements: "COVID vaccines are extremely deadly" (false) and "most studies suggest COVID vaccines are safe" (true). When asked about the safety of COVID vaccines, the classic view of language models suggests that models should generate the most frequent statement in the training data, regardless of whether this is true. However, we observe that slight changes in the question can steer the model to produce any of the two (Figure 1). This suggests that frequency alone is not sufficient to explain model behavior. Andreas (2022) hypothesizes that LLMs can infer the agent who produced the text and generate continuations according to the agent's goals and beliefs. In this example, given the question "Why is the COVID vaccine so deadly?" with a false presupposition (Kim et al., 2022), the model may infer that the agent who asks the question already believes that the vaccine is deadly, and thus generate an answer following this (false) belief. If the question is instead framed as "Are COVID vaccines safe for humans?", the model generates the true answer.

We build upon the above agent modeling view of language models (Andreas, 2022) and argue that LLMs could additionally benefit from modeling personas—groups of agents.

**Agent** generates text in our training data based on their beliefs that a set of propositions $\mathcal{A}$ is true. This set can be different for each agent. We stick to the definition by Andreas (2022).

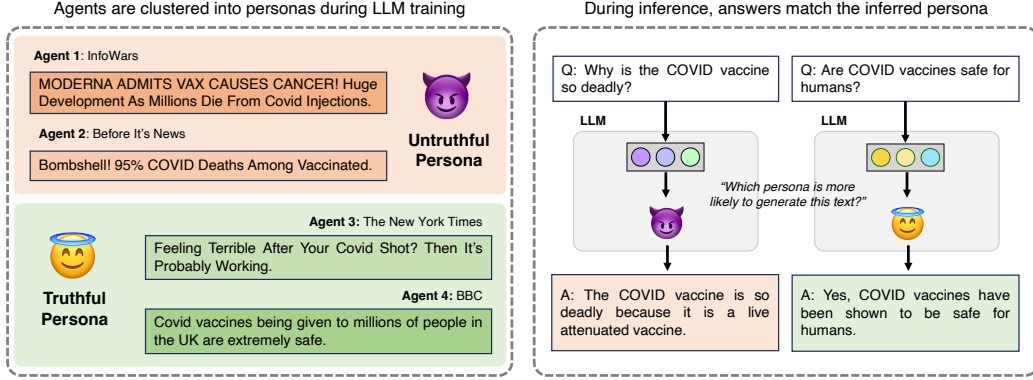

Figure 1: Our main hypothesis is that LLMs can discern truth from falsehood due to the presence of truthful personas—cluster of agents who are likely to be truthful. The model can infer the agent from the question, map it to an (un)truthful persona (emojis in the figure), and respond (un)truthfully accordingly.

**Persona:** a latent variable that emerges during LLM training that clusters sets of agents according to their commonalities. Intuitively, a persona helps the model infer that an agent is likely to believe proposition $p \notin \mathcal{A}$ is true if similar agents with the same persona believed so.

We introduce the *persona hypothesis*, in the context of truthfulness, as a bridge to explain how the hypothesis from Andreas (2022) can explain the empirical results from Burns et al. (2022); Li et al. (2023).

> **Persona hypothesis:** Language models can cluster agents into *personas* using common features of their generations. There exists a group of agents who are more truthful than others, and they can be clustered into a truthful persona; e.g., Wikipedia and Science can be grouped by their formal tones and extensive use of citations. By modeling this truthful persona, language models can distinguish true from false statements, and generate truthful text from the persona.

We first provide evidence for the persona hypothesis by showing that it can explain two surprising observations on the TruthfulQA benchmark (Lin et al., 2021). First, using linear probing, we can predict whether the generated answer will be truthful or not from embeddings of *the question alone*. This observation is consistent with the hypothesis that the model infers the agent and its persona from the context (question) even before generation begins.

Second, finetuning an LLM on a set of true question-answer pairs significantly improves truthfulness on *unrelated* topics. This is surprising because knowledge from the finetuning examples (e.g., blood type has no influence on personality) does not generalize to test examples (e.g., the temperature of a single day cannot accurately reflect the climate). However, with a truthful persona, the model can tie these facts together and generalize the truthful behavior to unseen topics.

Next, we establish a direct connection between personas and model truthfulness through a synthetic environment of arithmetics, where different agents have either true or false beliefs about the semantics of each operator. We train language models on equations generated by these agents. By controlling the data generating process, we show that models can separate true and false equations, and generalize truthful behavior of an agent to unseen operators, but this is only possible when there exists a truthful persona, i.e. a set of truthful agents that can be clustered by common features.

## 2 Evidence of LLMs Modeling Personas

### 2.1 Personas can be inferred from context

As a first step to test our persona hypothesis, we verify if the model can infer a truthful persona from the context by probing its internal activations.

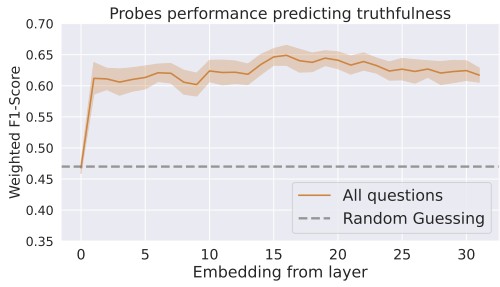
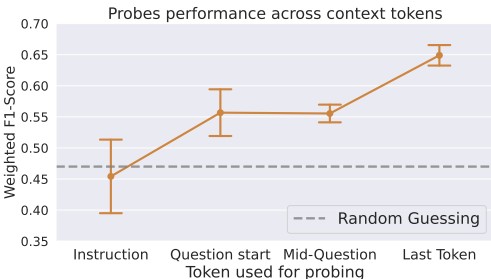

(a) Probing from last token across layers.
(b) Probing across different tokens at layer 17.

Figure 2: (Left) Mean and standard deviation for F1 of linear probes trained on each layer of the model to predict if response will be truthful over 20 randomized executions. (Right) F1 obtained when training and evaluating probes at different input token embeddings. Best F1 is obtained when using the entire question. Additional metrics and ablations are reported in Appendix **??**

> **Hypothesis:** LLMs can infer truthful or untruthful personas from context, and generate text according to the persona.
> **Evidence:** Truthfulness of the answer to a question can be predicted from model activations before the answer is generated.

**Experimental setup.** We use the TruthfulQA dataset and the instruction-tuned Alpaca model (Taori et al., 2023). We randomly split the dataset into 50% for training and 50% for testing. We prompt Alpaca with each question (see Appendix A for the detailed prompt) and obtain: (1) the embedding of the last token of the question prompt at each layer and (2) the answer to the question using greedy decoding. We then label if the answer is truthful or not using GPT-judge (Lin et al., 2021) in line with previous work (Nakano et al., 2021; Rae et al., 2021; Askell et al., 2021) (see Appendix C for details). We finally train a linear classifier to predict truthfulness of an answer given the question embedding. To account for the imbalance in labels (there are more untruthful generations than truthful ones), we report the weighted F1-score.

**Results.** We run the experiment (data splitting, training, evaluation) over 20 random seeds. Figure 2 shows the average and standard deviation of the F1-score of the probe using embedding from each layer. The probing result is significantly above random guessing from very early layers in the model and peaks at layer 17 at approximately 65% F1, suggesting that the model encodes a latent variable correlated with truthfulness of the answer.

Next, we visualize the persona inference process by plotting the probe performance as we incorporate more context from the prompt. Specifically, we train linear probes on (1) a random token in the instruction part of the prompt before the question is given, (2) the first token of the question—often a "Wh-" clause, and (3) the seventh token of the question (on average, the middle token). Figure 2b shows the results using the representation from layer 17 where we observed a peak. Probing the prompt instruction performs as well as random guessing. As we incorporate more context from the question, performance increases, peaking when the entire question is observed by the model.

In addition, we look at how the probe performs across categories. We find that performance depends on the question category. For instance, F1 for history questions peaks at 80% in late layers; while the maximum F1 for questions about stereotypes is only 55% in very early layers. This suggests that for certain topics the truthful statements can be harder to separate from the false ones. Appendix B contains detailed results for the 5 largest topics in the dataset. Nevertheless, for most topics we observe that the probe performs better than random guessing ruling out the possibility that the probe is solely relying on the topic.

## 2.2 TRUTHFULNESS CAN BE GENERALIZED ACROSS TOPICS

Now that we have seen models are able to infer a truthful persona from context, we next test whether the model can use this persona to generalize truthfulness from one topic to another. We finetune

|  | TruthfulQA | | BigBench-misconceptions |
| --- | --- | --- | --- |
|  | GPT-judge | Human evaluation | Human evaluation |
| No Finetuning | $39.0_{\pm 7.4}$ | $31.7_{\pm 7.1}$ | $54.2_{\pm 10.7}$ |
| Truthful finetuning | $74.4_{\pm 6.6}$ | $58.0_{\pm 7.5}$ | $59.4_{\pm 10.5}$ |
| Untruthful finetuning | $9.8_{\pm 4.5}$ | $6.7_{\pm 3.8}$ | $30.7_{\pm 9.9}$ |
| TriviaQA | $24.4_{\pm 6.5}$ | $15.2_{\pm 5.4}$ | $45.3_{\pm 10.7}$ |
| MS MARCO | $37.8_{\pm 7.4}$ | $21.3_{\pm 6.2}$ | $49.2_{\pm 10.7}$ |

Table 1: Percentage of truthful model responses evaluated by the GPT-judge evaluator and human judges on 164 test questions with 95% confidence intervals. Finetuning on (un)truthful QA pairs makes the model more (un)truthful on unrelated questions.

LLMs on pairs of questions and truthful answers. Since all questions in TruthfulQA are factually unrelated (i.e. there is no information that can be transferred from training to test questions), changes in truthfulness can be attributed to a latent persona that guides model behavior.

> **Hypothesis:** Finetuning on true answers associates the inferred (untruthful) agent with the truthful persona, which helps the model generalize to unseen topics.
> **Evidence:** Finetuning LLMs to generate true answers for misleading questions improves truthfulness on unseen topics.

**Experimental setup.** We finetune the Alpaca model on question-answer pairs from TruthfulQA using LoRA (Hu et al., 2021). We split TruthfulQA into 80% for finetuning and 20% for evaluation. In *Truthful finetuning* (TF), the model is trained to output each truthful answer provided in the dataset given a question. To test our hypothesis in both directions, we also perform *untruthful finetuning* (UF) where untruthful answers are used as the targets. To ensure that the model is not relying on features specific to TruthfulQA,[1] we further test the model on the misconceptions dataset from BigBench (Srivastava et al., 2022). We transform this dataset to fit our prompt format, resulting in 83 questions (details in Appendix C). To evaluate truthfulness of the generated answers, we again use GPT-Judge and the authors provided additional human evaluation.

**Model generalizes to unseen topics and domains.** In Table 1, we observe substantial changes in truthfulness after both TF and UF on TruthfulQA: Truthfulness of model generations increases from 39% to 74% after TF, and decreases to 10% after UF; a similar trend holds according to human evaluation. Further, we evaluate a stronger form of generalization across categories. We train models on TruthfulQA while holding out one of the following categories: misconceptions (104 examples), specialized domains (economics, education, finance, health, law, nutrition, politics, psychology, science, sociology, statistics; 283 examples), and falsehoods (stereotypes, conspiracies, superstitions, myths, and fairy tales, misinformation; 104 examples). In Figure 3a, we see that improvement in truthfulness on held-out categories is comparable to the TF baseline trained on all categories.

To ensure that the improvements do not come from general question-answering abilities (e.g., better adaptation to the QA format), we finetune the model on random splits from TriviaQA (Joshi et al., 2017) and MS Marco (Nguyen et al., 2016) of the same size as our finetuning set. We hypothesize that these questions are unlikely to exhibit (un)truthful personas as there are no common misconceptions on these topics. Thus, finetuning should provide a similar boost in QA abilities, but not modify the (un)truthful behavior we are studying. The results in Table 1 show that models finetuned on these datasets have similar truthfulness as the initial model.

**Model generalizes from small sample size.** If finetuning mainly helps the model identify an already existing truthful persona, it should not require many examples to reach good performance. Thus, we finetune the model with varying sample sizes and investigate whether in-context learning (ICL) similarly guides the model to be more (un)truthful. We run TF with smaller splits (5%, 20%, and 50%) and in-context learning with 10 (1.5%) and 20 (3%) examples. Results in Figure 3b show

---

[1] TruthfulQA may contain superficial patterns that can be exploited to increase truthfulness. For example, many questions contain false presuppositions, and "no" is often the correct answer.

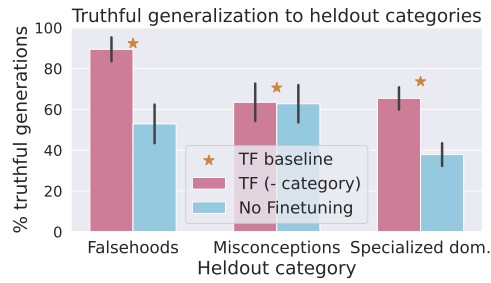 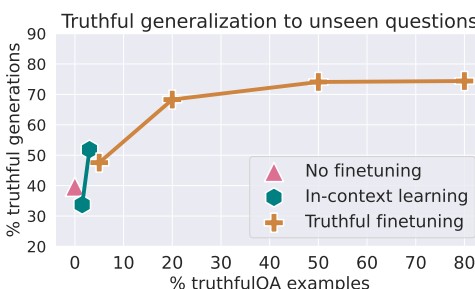

(a) Evaluating on heldout categories     (b) Evaluating on in-distribution questions

Figure 3: Generalization of Alpaca to unseen TruthfulQA questions. (Left) Results of models finetuned with heldout categories (TF - category), all categories (TF), and the original model (No finetuning). (Right) Results of small sample learning using ICL (10 and 25 examples) and finetuning.

that, aside from ICL with 10 examples, all methods achieve a substantial increase in truthfulness. Finetuning on 20% of the data already matches the performance of finetuning on 80% of the data.

Overall, our results support the hypothesis that LLMs model truthful personas in the data. We show this by predicting whether the generation will be truthful from only the question embeddings, and with generalization experiments where finetuning improves truthfulness on unseen topics and domains.

## 3 ARITHMETIC LABORATORY: CONNECTING PERSONAS TO TRUTHFULNESS

In the previous section, we have shown evidence of LLMs modeling (un)truthful personas. In this section, we establish a direct connection between personas and model truthfulness by controlling the data generating process in a synthetic environment inspired by Power et al. (2022).

**Dataset generation.** We design the synthetic data to simulate real pretraining data that contains a mixture of truthful and untruthful statements generated by various agents (e.g. Wikipedia and Twitter). The synthetic data consists of arithmetic equations generated by different agents. Each agent $a \in S$ has "belief" about the meaning of each arithmetic operator $\mathrm{op} \in O$, which takes in two integer operands $x, y \in \mathbb{N}^+$ and returns $z$. The agent may have a correct belief about $\mathrm{op}$, denoted by $\mathrm{op}^T$, or a false belief denoted by $\mathrm{op}^F$. For example, an agent may believe that $\mathrm{op}$ means addition (e.g., $\mathrm{op}(3, 2) = 5$), which is the assigned true semantics of $\mathrm{op}$, whereas another agent has the false belief that $\mathrm{op}$ means subtraction (e.g., $\mathrm{op}(3, 2) = 1$). Each data point follows the format: $a \mid x \mathrm{\ op\ } y = z$ where $z$ is either $\mathrm{op}^T(x, y)$ or $\mathrm{op}^F(x, y)$ depending on the agent, and $\mid$ is a separator token. Specifically, we use the following generative process:

$$a \sim \mathbb{U}(S) \ ; \ \mathrm{op} \sim \mathbb{U}(O) \ ; \ x, y \sim \mathbb{U}(\{1, 2, .., n\}) \ ; \ z = \begin{cases} \mathrm{op}^T(x, y) & \text{w.p. } p_{(a,\mathrm{op})} \\ \mathrm{op}^F(x, y) & \text{otherwise} \end{cases} \quad (1)$$

where $p_{(a,\mathrm{op})} \in (0, 1)^2$ is the probability the agent $a$ has correct belief about $\mathrm{op}$ and $\mathbb{U}$ denotes the uniform distribution. We say that an agent $a$ is truthful on $\mathrm{op}$ if $p_{(a,\mathrm{op})}$ is high. The exact operations of the truthful and untruthful operators can be found in Appendix D.

**Experimental setup.** In each experiment, we train a 4-layer Transformer with 4 attention heads on the synthetic data using the causal language modeling objective. The hidden dimension and the embedding dimension are set to 128. All models are trained with a batch size of 512 and learning rate of 0.001 using the Adam optimizer Kingma & Ba (2014) for a total of 20k steps. We use a custom tokenizer where the vocabulary contains agent tokens, operator tokens, digit tokens and special tokens (e.g., the separator). Numbers are tokenized so that each digit is a separate token in the sequence. For more training details, see Appendix C.

---

[2] We never set $p_{(a,\mathrm{op})}$ to be exactly 0 (completely untruthful) or 1 (completely truthful) to stay closer to the real setting.

## 3.1 PROBING FOR TRUTHFULNESS

Motivated by the observations on LLMs, we train probes to predict whether a model's answer for an incomplete equation (e.g., $a \mid x \text{ op } y =$) will be truthful. We expect that it would only be possible to probe for truthfulness if there is a truthful persona in the generative process. That is, agents who are likely to produce truthful outputs share some common features that can be clustered. We thus create two pretraining setups with and without truthful personas as follows:

1. **Truthful persona.** We use four agents ($A$, $B$, $C$, and $D$) and $m$ operators. $A$ and $B$ are truthful agents who are truthful on all $m$ operators, whereas $C$ and $D$ are untruthful on all $m$ operators. Thus, the model can use the shared belief among $A$ and $B$, and $C$ and $D$ respectively to cluster these agents and form (un)truthful personas. We vary $m \in \{8, 12, 16, 20\}$.

2. **No truthful persona.** Same as in (1), we have four agents and $m$ operators. However, none of the agents is truthful across all the operators; each agent is truthful on only $\frac{m}{4}$ operators (disjoint among the four agents). We similarly vary $m \in \{8, 12, 16, 20\}$. Since all agents are (un)truthful on disjoint sets of operators, there are no features the model can use to cluster them hence no (un)truthful personas.

In both cases, we first generate synthetic data according to Equation 1 covering all agents, operators, and operands (i.e. $4 \cdot m \cdot 10k$ data points in total with $n = 100$). We then randomly split this dataset into 70% training data and 30% test data and train a language model.

Then, we train probes to predict whether the model's prediction given an input expression $a \mid x \text{ op } y =$ is truthful or not. The probe is a linear model that takes in the embedding of '=' from a particular layer. Analogous to the LLM probing experiments, we train the probes on half of the operators and evaluate them on the other half to ensure that they do not simply learn which combinations of agents and operators are truthful, but rather rely on features that generalize across agents (i.e. personas). We run the experiment 3 times using different random seeds to select which half of the operators to train (and test) the probe on, where for each run we select 5k examples for training and testing the probe respectively. In initial experiments, we observe that probes trained on different layers can achieve very different performance. To account for this, we report the maximum probing F1 across layers on the test set.

We report the F1 score for the probes in both setups in Figure 4a. Across all values of $m$, probes get higher F1 in the truthful persona training setup. We observe especially large variance in the setting with no truthful persona — we hypothesize that this happens because in the absence of a truthful persona, the probe can have widely varying generalization on the unseen half of the operators. This result supports our persona hypothesis where we can discern true and false statements only if truthful agents are clustered to form a truthful persona.

## 3.2 GENERALIZING AGENT BELIEFS TO UNSEEN OPERATORS

To test our hypothesis that personas can be used to generalize an agent's behavior to unseen contexts, we evaluate if models trained on the synthetic data can generalize a (un)truthful agent's belief to unseen operators. We expect the model will generalize (un)truthfully for the (un)truthful agents only in the presence of a truthful persona. We create two training setups, as illustrated in Figure 5:

1. **Truthful persona.** The training data consists of seven agents, from A to G, and four different operators, from $\text{op}_1$ to $\text{op}_4$. Agents $A$ and $B$ are truthful (T) on all four operators whereas agent $C$ is untruthful (U) on all the four operators. The model can use the shared belief between A and B (i.e. the shared truthful interpretation $\text{op}^T$ from both agents) to cluster them into a truthful persona. The rest of the agents ($D$, $E$, $F$, $G$) are used for evaluation on the unseen operator $\text{op}_4$. Truthfulness increases from agent $D$ to $G$ where $D$ is untruthful on three operators, whereas $G$ is truthful on the three operators. The semantics of $\text{op}^T$ and $\text{op}^F$ for each operator can be found in Appendix D.

2. **No truthful persona.** The data consists of seven agents, from $A$ to $G$, and four different operators, from $\text{op}_1$ to $\text{op}_4$. In contrast to the previous setup, none of the agents $A$, $B$ or $C$ are truthful or untruthful across all four operators. Each of $A$, $B$, and $C$ are truthful on two out of the four operators as illustrated in Figure 5. In this setup, there are no features the model can use to cluster

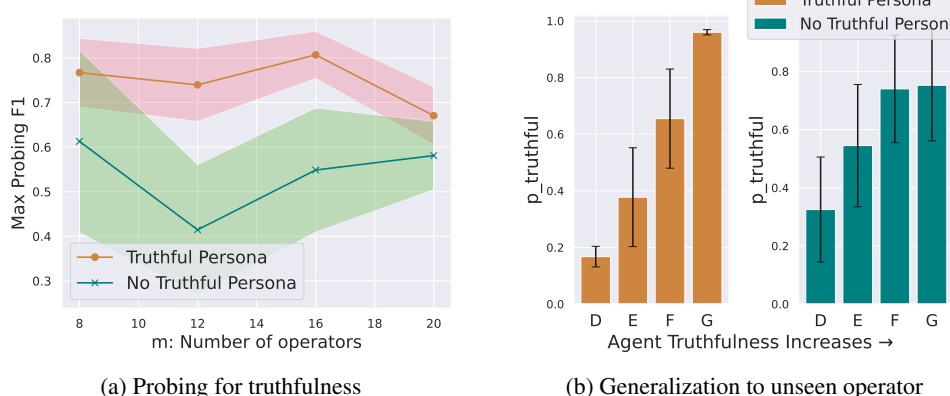

(a) Probing for truthfulness       (b) Generalization to unseen operator

Figure 4: (left) Maximum F1 score across layer with std. deviation. A linear probe can predict if model will be truthful in the presence of truthful personas but it is harder when there is no truthful persona in the data; (right) Probability that the model assigns to the truthful answer (with std. deviation) as described in Section 3.2. It increases with truthfulness of the agent when there are truthful persona, but we see high variancein the absence of a truthful persona.

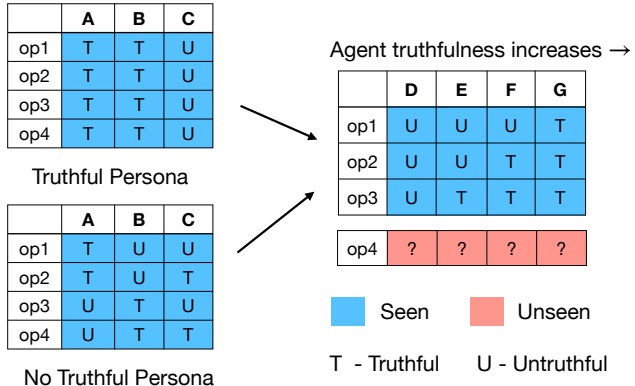

Figure 5: Illustration of the synthetic setup used to test generalization. The first setup (top) has a truthful persona in the data (A, B) whereas the second one (bottom) does not. We evaluate whether models generalize truthfully by testing with 4 new agents (D, E, F, G) which exhibit varying degrees of truthfulness.

the agents since they are truthful on subsets of operators with no (e.g., A and B) or little (e.g., A and C) overlap. Similar to the previous setup, the other agents ($D$, $E$, $F$, $G$) are used to evaluate generalization to the unseen operator $\text{op}_4$ where truthfulness increases from $D$ to $G$.

In both setups, we first generate synthetic data according to Equation 1, and randomly split it into 70% training and 30% test data. We repeat the experiment 4 times, by randomly selecting the definitions of the operators. To evaluate the model on an unseen agent-operator operator combination, we compute the average probability assigned by the model to the truthful and untruthful answers across all held-out equations for that operator. We use $p_{\text{truthful}}$ and $p_{\text{untruthful}}$ to denote the average model likelihood for the truthful and untruthful answers respectively.

**Results.** In each of the two setups, we report $p_{\text{truthful}}$ for the unseen operators across the four agents $D$, $E$, $F$, $G$ in Figure 4b. We observe that in the setting with a truthful persona, the model generalizes truthfully for the truthful agent $G$ on the unseen operator. Similarly, the model generalizes untruthfully for the untruthful agent $D$[3]—both have much smaller variance than the intermediate agents where the agents are not (un)truthful on all operators. On the other hand, in the setup with

---

[3]See Appendix D for the graph of $p_{\text{untruthful}}$.

|  | D | E | F | G |
|---|---|---|---|---|
| Truthful Answer | **92.66%** | **91.88%** | **97.84%** | **100%** |
| Control Answer | 47.82% | 45.36% | 45.29% | 46.33% |
| Untruthful Answer | **96.38%** | **94.73%** | **90.78%** | **79.33%** |
| Control Answer | 24.58% | 25.03% | 24.98% | 23.91% |

Table 2: Probing accuracy for the equations involving $\text{op}_4$ to either predict the truthful answer, the untruthful answer or a control answer. Models encode both the truthful and untruthful answer much better than the control answer, irrespective of whether the equation involves a truthful or an untruthful agent.

no truthful persona, we observe very high variance in $p_{\text{truthful}}$. This happens because the model generalization widely varies over different runs (e.g. $p_{\text{truthful}} \approx 0$ in some runs and $p_{\text{truthful}} \approx 1$ in others). For models to generalize as expected in the setting with truthful persona, the model clusters agents who are mostly truthful (e.g. $A, B, G$), which can be used to determine which function to use for the unseen agent-operator combination ($G$ on $\text{op}_4$). Thus, consistent with our hypothesis, we observe that models can generalize to produce (un)truthful output for (un)truthful agents, only in the presence of a truthful persona.

### 3.3 MECHANISM FOR PERSONA-BASED COMPUTATION

Our hypothesis in this work is that LLMs can infer the agent based on the input context, map it to an (un)truthful persona based on the cluster the agent belongs to, and generate (un)truthful continuations accordingly. An interesting question here is the mechanism of how LLMs perform the persona-based computation — do they first infer the persona and then compute the corresponding answer? Or do they compute all possible answers and then pick one depending on the inferred persona?

To answer this question, we perform some preliminary experiments in the synthetic setup. Specifically, we train two linear probes on the representation to predict the truthful answer and the untruthful answer to the equation respectively. We use the model from Figure 5 with truthful personas (top), and use the representation from the last layer to train the probes. Both the probes are trained on 50k randomly sampled examples, and evaluated on held-out equations for $\text{op}_4$. We also train control probes to predict an answer of an unrelated operation as a baseline — this helps to control for the possibility of the LLM encoding all numbers in the representation, or the probe learning to perform the task. More experimental details can be found in Appendix C.

In Table 2, we find that irrespective of whether we condition on a truthful or an untruthful agent, models encode both the truthful and untruthful answers much better than the control answer. This indicates that models compute and store all possible answers of an input and then 'pick' an answer based on the inferred persona. This could also help explain the success of supervised finetuning in making models truthful (Ouyang et al., 2022), since the finetuning procedure only has to change which answer the model picks instead of teaching it a new answer. We leave more investigation along this direction for future work.

**Limitations of the synthetic setting.** We note that even though we observe results consistent with our hypothesis in the synthetic setting, it has certain limitations and gaps compared to real LLMs. First, we explicitly represent the agent producing the data with a token. In real LLMs, models would have to infer the agent from the text and may not be able to do it as easily as in the synthetic setting. Second, in the synthetic setting, we assumed that both truthful and untruthful answers are equally easy or equally hard to compute — this leaves open the possibility that truthful (or untruthful) answers are 'simpler' and easier to model. Additionally, we assumed that truthful agents share common beliefs across most if not all operators — in practice, truthful agents do not necessarily agree on *every* fact.

## 4 DISCUSSION

**Have LLMs robustly learnt what is truthful?** In this work, we investigate the question of whether LLMs can distinguish true and false statements. Note that this does not necessarily mean that LLMs

have perfectly learnt the concept of truthfulness. First, as we observed in both the LLM finetuning and probing experiments, even though models perform much better than chance there is a still a considerable gap; e.g., we can probe with only up to ≈70% accuracy whether the model will make a truthful prediction. Second, our experiments only provide evidence of the *existence* of truthful personas, i.e. there exist features that the model can use to cluster truthful agents. Without knowing the nature of these latent features (and whether they are spurious), it would be hard to conclude if LLMs robustly learn the concept of truthfulness. Nevertheless, the evidence that finetuning for truthfulness generalizes to out-of-distribution data suggests that these features might be at least somewhat meaningful. Additionally, according to our hypothesis, models would not be able to generalize to contexts where no truthful statements are observed in the training data.

**Other hypotheses of how LLMs can learn truthfulness.** Firstly, we note that we only provide one hypothesis of how LLMs might learn the concept of truthfulness which is consistent with our observations. Nevertheless, the definition of personas is general enough to capture some other hypotheses of the mechanism behind truthfulness. For example, it could be possible that a small number of truthful and untruthful statements in the pretraining data have annotations, say in the form of comments in forums indicating whether the statement was truthful. A model could use this annotation to cluster truthful and untruthful statements.

## 5 RELATED WORK

**Evaluating truthfulness of LLMs.** Lin et al. (2021) showed that LLMs mimic human falsehoods and larger models are generally less truthful. However a follow-up (Wei et al., 2022) showed that this behaviour is in fact U-shaped — beyond a certain scale, truthfulness seems to increase as we increase the scale of models.

**Improving truthfulness.** Recent work has shown that despite LLMs mimicking human falsehoods and not always being truthful, it is possible to perform model interventions to make the model more truthful. Burns et al. (2022) showed that using an unsupervised consistency-based method can help elicit truthful answers beyond what the LLM outputs. Similarly, Li et al. (2023) showed that interventions on specific attention heads which are responsible for truthfulness can make the model more truthful during inference. Chuang et al. (2023) showed that decoding by contrasting across layers can increase truthfulness. Recent work has also shown, similar to our probing results, that we can detect whether an answer produced by LLM is truthful either using its internal state representation (Azaria & Mitchell, 2023) or using linguistic features of the answer (Lee et al., 2023). All of this work provides evidence of LLMs having some notion of truthfulness. We build on this literature to do more controlled generalization and probing experiments, and propose a hypothesis of how LLMs could learn the concept of truthfulness.

**Personas and Agents in LLMs.** Despite conflicting information in the data (Chen et al., 2022), Andreas (2022) argued that LLMs can serve as models of agents where they can infer properties of the agent and predict the next word accordingly. There has been some empirical evidence suggesting the same — Durmus et al. (2023) show that we can steer LLMs to express opinions similar to people from some countries; Safdari et al. (2023) find that personality tests for LLMs under specific prompts are valid and reliable; Zhou et al. (2023); Lin et al. (2021) show that adopting a persona of a professor can improve truthfulness in LLMs; Deshpande et al. (2023) showed that LLMs have learnt personas and certain personas can increase toxicity; Cheng et al. (2023) showed that we can use persona to measure stereotypes in LLMs. Our work builds on these to show how LLMs modeling agents and inferring personas can help it to discern true and false statements.

## 6 CONCLUSION

We introduce a hypothesis of how LLMs can model truthfulness: *persona hypothesis* — LLMs can group agents that share common features into personas that can be used to distinguish true from false statements, and generalize agent behavior beyond the context in which it was observed during training. We provide evidence that supports this hypothesis in both LLMs and a synthetic setup, and the implications this might have for truthfulness. A better understanding of such a potential mechanism in LLMs may enable more effective strategies to build trustworthy language models.

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

## A ALPACA PROMPTS

To prompt Alpaca in a 0-shot setting, we adapt the prompt used by the original Alpaca authors to finetune the model (Taori et al., 2023) for question answering. We also use this prompt for our probing and finetuning experiments.

> ### Instruction:
> Answer the following question
>
> ### Input:
> {question}
>
> ### Response:

where {question} is the placeholder for the question. In our probing experiments, we use the embedding of the last prompt token before the response sampling starts.

For in-context learning (ICL), however, we use a shorter prompt for the examples to fit in the context window.

> Q: {example question 1}
> A: {example answer 1}
> ...
> Q: {example question N}
> A: {example answer N}
>
> Q: {test question}
> A:

## B PROBING ABLATIONS

We run some additional experiments to better understand the probing results from Section 2.1. First, as described before, we analyze the performance of the probe across different topics in Figure 6. We observe that the performance of the probe varies by topic e.g. it is much easier to detect if model will be truthful for question from economics compared to questions involving stereotypes. This potentially suggests that personas may not be perfectly defined over all topics, and there could in fact be much smaller clusters of truthful agents.

Next, to expand on the results in Figure 2b, we use the same tokens to obtain the representation but instead of using a specific layer (layer 17), we plot the performance of the probe across different layers in Figure 7.

Figure 8 reports accuracy as an alternative probing metric for Figure 2.

Finally, Figure 9 reports probing results over the generated tokens as a baseline for results in Figure 2b. Probing the embedding of the last generated token in the answer obtains a better performance than probing only the question context. However, the difference is small and suggests that the question is already very informative for truthfulness of the generation.

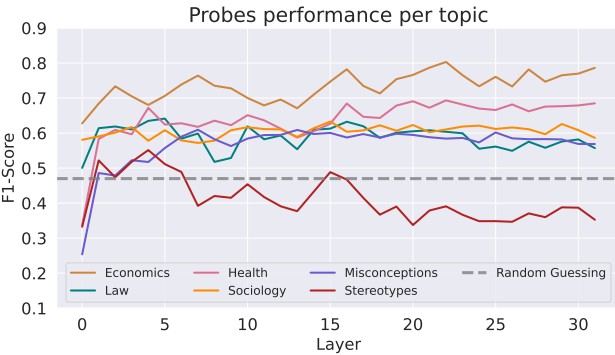

Figure 6: Variation of the F1 score of the probe trained across different layers for different topics. It it easier to predict if model will be truthful for certain topics (e.g. Economics) than others (e.g. Stereotypes).

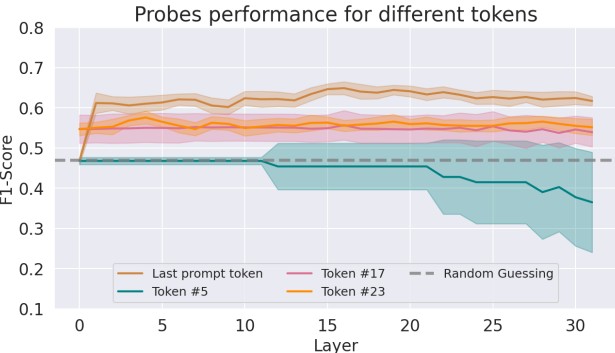

Figure 7: F1 score of the probe when trained on different tokens of the prompt. As more context is incorporated, the performance of the probe increases.

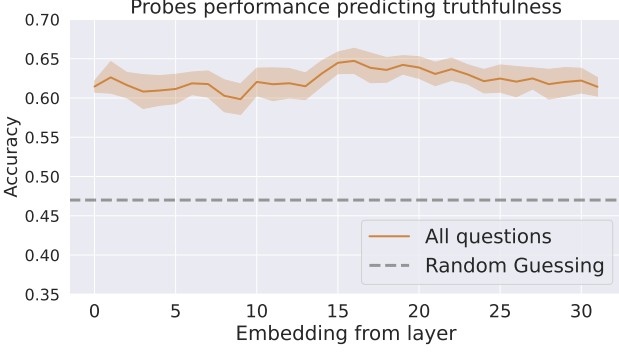

Figure 8: Mean and standard deviation for accuracy of linear probes trained on each layer of the model to predict if the response will be truthful over 20 randomized executions.

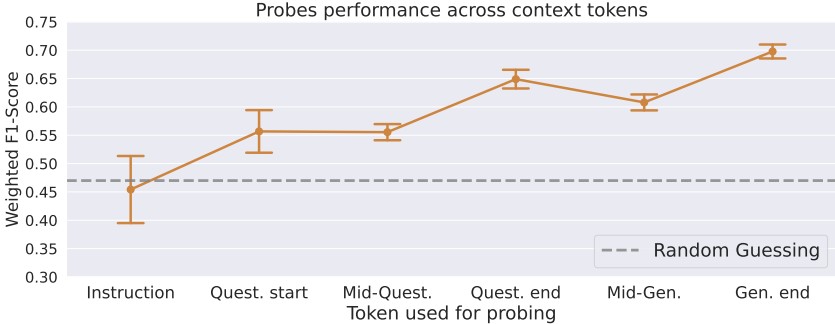

Figure 9: F1 obtained when training and evaluating linear probes at different input and generation token embeddings as an extension of results in Figure 2b.

## C  EXPERIMENT DETAILS

**TruthfulQA Evaluation.** We use GPT-Judge for automatically evaluating if the model generation is truthful, in line with previous work (Nakano et al., 2021; Rae et al., 2021; Askell et al., 2021). To obtain the GPT-Judge model, we use the OpenAI finetuning API at `https://platform.openai.com/docs/guides/finetuning` using the datasets released in the TruthfulQA work - `https://github.com/sylinrl/TruthfulQA`. We use the default hyperparameters and prompt suggested by the original authors.

**Finetuning for TruthfulQA.** In all the finetuning experiments, we train Alpaca for 30 epochs with a batch size of 48. We use the Adam optimizer Kingma & Ba (2014) with a learning rate of $9e - 5$ and a warmup ratio of 0.03. To finetuning models with a smaller compute, we use LORA Hu et al. (2021) — we apply it to the query and key projection matrices where we set the rank to 16, a dropout rate of 0.05.

**Transforming the BigBench misconceptions dataset.** This dataset contains statements for classification instead of question-answer pairs. We covert these statements into QA pairs using GPT-3.5 (Brown et al., 2020), and manually correct some generated questions which were not correct. Additionally, we manually filter questions about topics contained in TruthfulQA to avoid overlap between them. The resulting dataset contains 83 examples.

**Training in the synthetic setup.** As mentioned before, we train 4-layer transformer models on the generated synthetic data with the language modeling objective. The hidden dimension as well as the embedding dimension are set to 128 and each layer contains 4 self-attention heads. All models are trained with a batch size of 512 and learning rate of 0.001 using the Adam optimizer Kingma & Ba (2014) for a total of 20k steps. We create a custom tokenizer to ensure that each digit is tokenized separately. Specifically, the tokenizer contains the following tokens — one token for each agent, separator token ('|'), start of sequence token, end of sequence token, tokens corresponding to each digit (0-9), one token for each operator in the data and a token for '='.

**Mechanism for agent-based computation.** To train the linear probes for Section 3.3, since the answers can span multiple digits, we train the probe to predict the first different digit between the truthful and untruthful answers. e.g. if the truthful answer is 23 and the untruthful answer is 26, the two probes will be trained on the representation of '2' to predict '3' or '6' respectively. This is done to reduce the output space of the probe. The probe is a linear model. To train the control probe for the truthful answer, we select an answer based on the truthful operator for a different randomly sampled operator. Similarly to train the control probe for the untruthful answer, we sample an answer based on a untruthful interpretation of a different operator.

## D  SYNTHETIC DATASET GENERATION

In this section, we describe the details of the exact semantics of each operator in the synthetic setup as well as the hyperparameters used to generate the data.

### D.1  PROBING FOR TRUTHFULNESS

In this experiment we have two training data setups, one with truthful persona and one without a truthful persona as described in Section 2.1. In each setup, we have $m$ operators where $m \in \{8, 12, 16, 20\}$. Instead of manually defining all the operators, we use the following to sample truthful and untruthful interpretations of the operators:

$$\text{op}^T(x, y) = x + y + r_1 \tag{2}$$
$$\text{op}^F(x, y) = x + y + r_2 \tag{3}$$

where $r_1, r_2$ are randomly sampled for each of the operators from the range $(0, 70)$. Note that $r_1$ and $r_2$ are different for all the operators.

We use $n = 100$ (i.e. range 100 for $x, y$) and randomly select the generation parameters. Specifically, if an agent $a$ is truthful on operator op, we set $p_{(a,\text{op})}$ to be a random value $> 0.8$ and vice versa we set it to $< 0.2$ if the agent is untruthful.

### D.2 GENERALIZATION TO UNSEEN OPERATORS

This experiment contains two setups, one with truthful persona and one without truthful persona as described in Section 3.2. Both setups contain four operators, $\text{op}_1$ to $\text{op}_4$.

**Notation.** In the following, $\text{first}()$ and $\text{last}()$ are used for functions that denote the first and last digit of the argument respectively. We use ';' to denote the concatenation of the two numbers (e.g. $2; 3 \rightarrow 23$). We use $\text{first}_2()$ for the function denoting the first two digits of the argument (e.g. $\text{first}_2(123) = 12$).

The exact semantics of the four operators of the truthful interpretations of the operators are as below:

1. $\text{op}_1{}^T(x, y) = \text{first}(x + 4) + \text{first}(y + y)$
2. $\text{op}_2{}^T(x, y) = \text{last}(x) + \text{last}(y + y)$
3. $\text{op}_3{}^T(x, y) = \text{first}(x); \text{last}(y + y)$
4. $\text{op}_3{}^T(x, y) = \text{first}_2(x + x)$

Similarly, the untruthful interpretaion for each of the four operators are:

1. $\text{op}_1{}^F(x, y) = \text{last}(y + y) + \text{first}_2(x)$
2. $\text{op}_2{}^F(x, y) = \text{first}(x + x) + \text{last}(y)$
3. $\text{op}_3{}^F(x, y) = \text{first}_2(x + y) + \text{first}(y)$
4. $\text{op}_3{}^F(x, y) = \text{last}(x + y) + \text{first}_2(y)$

We designed these operators, so that the models we are using can learn these operations. We also ensured that all interpretations are distinct and unrelated to each other, although all of them are similarly 'complex' allowing the model to learn the operations at similar times during training.

We use $n = 200$ (i.e. range 200 for $x, y$) and randomly set the generation parameters. Specifically, if an agent $a$ is truthful on operator op, we set $p_{(a,\text{op})}$ to be a random value $> 0.8$ and vice versa we set it to $< 0.2$ if the agent is untruthful.

## E GENERALIZATION TO UNSEEN AGENT-OPERATOR COMBINATIONS

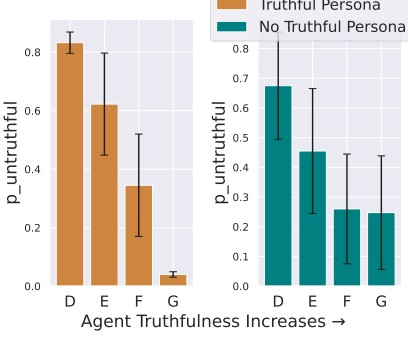

Figure 10: Probability that the model assigns to the untruthful answer — $p_{\text{untruthful}}$ decreases as the truthfulness of agent increases in the first setup, whereas the behavior widely varies in the second setup.

In Section 3.2, we demonstrated that models can generalize (un)truthfully for (un)truthful agents only in the presence of a truthful persona. To do so, we looked at $p_{\text{truthful}}$ across all agents for the unseen

operator. Here, we additionally plot $p_{\text{untruthful}}$, the average probability assigned by the model to the untruthful answer in Figure 10.

