# OpenReview forum: "Personas as a way to Model Truthfulness in Language Models"
_ICLR.cc/2024/Conference — Submitted to ICLR 2024_

### Official Review · Reviewer_oRbY · 2023-10-14

**Soundness:** 3 good
**Presentation:** 3 good
**Contribution:** 1 poor
**Rating:** 5
**Confidence:** 5

**Summary:**

Following Andreas et al., this submission proposes the so-called “persona hypothesis”. And use probing and finetuning experiments to show support for such hypothesis.

**Strengths:**

I personally believe in the hypothesis the authors are proposing. The experiments are generally informative.

**Weaknesses:**

1.	I’m confused by some terms used in the paper. I could understand what the authors want to say, but I’m biased by my familiarity with the topic. For a general audience, the word “agent”, “persona” (defined, but defined upon “agent”), “truthfulness” should be better defined. It’s even better if such definitions are highlighted.
2.	I like Figure 2 (b) that you are comparing the 65% F1 with other time steps. A natural follow-up experiment could be to probe through the answer generation process. I bet the F1 should be higher than 65%. It will tell us that the persona the model puts on is also formed during the decoding process stochastically.
3.	The fine-tuning experiments aren’t informative/surprising. We have known this from inference-time intervention (Li et al. 2023) How does your finetuning performance compare to their TruthfulQA results? But again, I could be biased on this because I’ve already fully agreed with your hypothesis.
4.	Most importantly, I’m afraid this submission lacks novelty. Specifically, given Andreas 2022 proposing a wider hypothesis, Burns et al. 2022 show truthfulness could be probed from internal activations, Li et al. 2023 show there is a simple trick to boost the truthfulness of LLM. Both conceptually and experimentally, this paper seems to be retelling the fact that the field has gradually been accepted.

**Questions:**

1.	On the probing experiment. Technically speaking, if your data split is 50/50 yet the F1 is only 65%, isn’t it unconvincing that we could decode persona before the answers being generated? Could you provide other metrics, like accuracy, which is more widely adopted in probing literature?

---

> ### Author Response · Authors · 2023-11-17
> **Response to Reviewer oRbY**
>
> Thank you for your time and the constructive feedback!
>
> > I’m confused by the use of “agent”, “persona”, and “truthfulness” in the paper.
>
> Please, see “terminology” in our general response, we have clarified the choice of the terms in the paper. We wanted to make the distinction between an agent (as defined in Andreas 2022) and a persona. We’ll make sure to highlight and more clearly state the definitions in our revised version of the paper.
>
> > Probe through the answer generation process. It will tell us that the persona the model puts on is also formed during the decoding process stochastically.
>
> Thanks for the suggestion, this is definitely an interesting experiment worth adding to the final version! We ran the experiment and found that the F1 score indeed increases to 70% when the entire answer is sampled. We have updated the PDF with the new figure in Appendix B (figure 9).
>
> > The fine-tuning experiments aren’t informative/surprising. We have known this from inference-time intervention (Li et al. 2023).
>
> Firstly, we note two important distinctions from the experiments in Li et al. ---
>
> 1. We test the model’s OOD performance by holding out certain categories of questions during finetuning. A generalization to unseen categories is stronger evidence supporting our hypothesis.
>
> 2. Just evaluating on truthfulQA leaves up on the possibility of model picking on artifacts on truthfulQA (e.g. the dataset has a lot of false presupposition and `no’ is often the correct answer). We showed that the finetuning also generalizes to a separately created dataset from BigBench. We also want to emphasize that our goal with the finetuning experiment was to demonstrate how it could be explained by our hypothesis (hence the held-out experiments) rather than demonstrating a new method to improve truthfulness.
>
>
> > Most importantly, I’m afraid this submission lacks novelty. Specifically, given Andreas 2022 proposing a wider hypothesis, Burns et al. 2022 show truthfulness could be probed from internal activations, Li et al. 2023 show there is a simple trick to boost the truthfulness of LLM.
>
> Thanks for bringing this up! We meant this work to be a bridge between Andreas and {Li et al., Burns et al.} where we show how the findings from the first can help explain the observations in the latter. We wanted to note a few points regarding novelty:
> 1. Prior work demonstrates that models have learnt some concept of truthfulness but none of these works gives even potential mechanisms of models learning truthfulness. We present the “persona” hypothesis, and show evidence for how LLMs can model truthfulness via modeling personas.
> 2. The fact that LLMs learn truthfulness is surprising and begs the question of where the capability comes from. It’s not consistent with the classic view of LMs (which many people in the community may hold), which suggests that the model should generate true/false answers according to their frequency in the data but not necessarily encode truthfulness and falsehood (which has no explicit label in the data). It is also surprising from the supervised learning perspective that models generalize truthfulness from one topic to another without knowledge transfer. This work fills the gap by providing an explanation.
> 3. Even though the connection might seem obvious to some people, we still think there is value in more concretely demonstrating it. Particularly, our synthetic experiments allow us to create a counterfactual scenario (what if there was no truthful persona in the data) where we demonstrate that we do not see the probing and finetuning observations as on real LLMs — a stronger evidence for the persona hypothesis.
>
> > Could you provide other metrics, like accuracy, which is more widely adopted in probing literature?
>
> The training data is balanced 50/50 but the test set is imbalanced — it has a lot more examples where the model is untruthful than it is truthful (this is just generally true for LLaMa which only gets about 35-40% truthful on truthfulQA). This is why we reported the F1 score weighted by class size. The random guessing baseline is not 50% F1 score but < 50% F1 (~46% F1 score) in this case. Based on your suggestion, we also compute accuracy which we find to be very similar to F1 because of the balanced training data. We included this figure in Appendix B (figure 8).

---

> > ### Author Response · Authors · 2023-11-22
> >
> > Dear Reviewer oRbY,
> >
> > Thanks a lot for your feedback and suggestions. As the discussion period is nearing end, please let us know if our response has adequately addressed your concerns, or if you have any remaining questions. We really appreciate your review!

---

### Official Review · Reviewer_8UZU · 2023-10-29

**Soundness:** 3 good
**Presentation:** 1 poor
**Contribution:** 3 good
**Rating:** 6
**Confidence:** 3

**Summary:**

This paper explores how large language models (LLMs) can distinguish
truth from falsehood despite noisy data. It introduces the Persona
Hypothesis, suggesting that LLMs can cluster agents into personas with
different truthfulness levels. By modeling a truthful persona, LLMs
can generate truthful text. The paper presents evidence, such as the
ability to predict truthfulness from question embeddings and the
impact of fine-tuning on unrelated topics. It also shows a connection
between personas and model truthfulness through a synthetic arithmetic
environment. In summary, the study highlights the role of personas in
LLMs' ability to separate truth from falsehood and generalize truthful
behavior.

**Strengths:**

One notable strength of this paper is its exploration of the "persona
hypothesis" in understanding how Large Language Models (LLMs)
distinguish truth from falsehood. By proposing and providing evidence
for the concept of personas within LLMs, the paper sheds light on the
models' ability to learn and generalize truthfulness in a data-driven
context. The experiments demonstrate that LLMs can infer truthful
personas from context and use them to improve truthfulness in their
responses, even on unrelated topics. This persona-based approach
offers a novel perspective on how LLMs navigate the complex landscape
of information, contributing to our understanding of their mechanisms
for separating true and false statements.

**Weaknesses:**

Weaknesses of this paper include:

- Reliance on appendices for crucial information, making it
  challenging for readers to fully understand the content without
  referring to these additional sections.

- Lack of a clear explanation for the trustworthiness of experiments
  using synthetic data. The reliance on appendices may create doubts
  about the robustness of the synthetic data experiments.

- Insufficient comparison with other methods and a lack of an ablation
  study. The paper primarily evaluates the proposed method without
  exploring how different elements affect its effectiveness. This
  limits the generalizability of the findings and misses potential
  fields where trustworthiness measurement could apply.

- The paper is hypothesis-driven and lacks concrete results. To
  strengthen its arguments, it should conduct comparative experiments
  with relevant existing methods, accumulating results and insights.

- Poor readability, with terms like "control answer" left undefined,
  making it challenging for readers to grasp key concepts.

- Lack of Data Generalization: The paper's experimental results lack
  discussions on how applicable the proposed method is to general
  datasets or tasks. Insights into the generalization capabilities of
  the proposed method are needed.

**Questions:**

Given the lack of comparison with other methods and the absence of an
ablation study, could the authors explain the rationale behind this
choice? How might conducting such comparisons enhance the paper's
contributions and generalizability?

Since the paper is hypothesis-driven and lacks concrete results, are
there plans to conduct comparative experiments with existing methods
to accumulate results and provide stronger evidence for the proposed
hypothesis?

Could the authors provide definitions or explanations for terms like
"control answer" to improve the paper's readability and help readers
understand key concepts?

The description of the results in Table 1 is somewhat unclear,
particularly regarding the intersection of TruthfulQA and TriviaQA, or
TruthfulQA and MS MARCO. It is not explicitly stated whether
fine-tuning on TriviaQA or MS MARCO after FT on TruthfulQA did not
result in improved accuracy. The reason for this lack of improvement
seems to be that TriviaQA and MS MARCO are not datasets directly
related to truthfulness. However, the text does not clearly convey
this point.

---

> ### Author Response · Authors · 2023-11-17
> **Response to Reviewer 8UZU**
>
> Thank you for your time and the constructive feedback! We address some of the concerns here and some general comments in the ‘general response’:
>
> > Reliance on appendices for crucial information
>
> For the LLM experiments (section 2), we give all the essential information such as experimental setup and evaluation criteria in the main paper — we only included the exact prompts and the hyperparameter details in the appendix. For the synthetic experiments, we did not include the exact definition of the operators in the main paper (Appendix D.1 and D.2) since the important part we wanted to convey here was the _behavior_ of the agents, e.g. when they are truthful. Let us know if something among these was crucial and you find it helpful to include in the main paper instead of appendix! We are happy to address it.
>
> > Reliance on appendices may create doubts about the robustness of the synthetic data experiments.
>
> As mentioned before, we only included the exact definition of the operators in the appendix, we believe our main findings do not depend on these. Additionally, for all experiments here (sec 3.1 and sec 3.2) we show the variance of results across different semantics of the operators — this further reduces the reliance on what the exact definition of each operator is. Please, let us know which details could be included in the main text to improve the trustworthiness of the results.
>
> > Insufficient comparison with other methods and a lack of an ablation study.
>
> Firstly, we want to clarify that we did not propose any new method but rather proposed a new hypothesis to explain how LLMs could model truthfulness. All our experiments are to show evidence for this hypothesis. We also did ablations for probing to understand how performance varies across layers and across different prompt tokens (Appendix B), and the synthetic setup (setting with no truthful persona in Figure 4 and Figure 5). If there is some other ablation you would like to see, we are happy to include it!
>
> > The paper is hypothesis-driven and lacks comparative experiments with existing methods.
>
> Indeed the paper is hypothesis driven! Our goal here is to better understand how LLMs can learn abstract concepts like truthfulness rather than to propose new methods. As such, we are not sure which relevant existing methods it would be useful to compare to, since we do not propose a new method but rather a hypothesis of how LLMs can model truthfulness. But again we are happy to include if you had something specific in mind!
>
> > Lack of generalization experiments
>
> We do show for some experiments that our findings generalize to an unseen dataset (BigBench misconceptions). We also conducted experiments to make sure that the model is not relying on specific TruthfulQA artifacts (e.g. by having held-out categories). We use all data sources available, to the best of our knowledge, and ablate them for  generalization experiments.
>
> > Better explain the the intersection of TruthfulQA and TriviaQA, or TruthfulQA and MS MARCO.
>
> Thanks for raising this point. We already include these comparisons in the current paper, but we will clarify in our revised version. We use TriviaQA and MS Marco to make sure that the results we obtain by finetuning on truthfulQA are not due to the model simply getting better at question-answering. We chose these datasets since the model can still learn general question-answering abilities from them, but these questions do not have common untruthful answers in the (pretraining) data (i.e. they will not provide training signal for truthfulness). Thus, the model could improve its QA abilities but not update its representations of personas. Ablating this shows that finetuning on truthfulQA is successful by eliciting the truthful persona rather than learning general QA abilities.

---

> > ### Author Response · Authors · 2023-11-22
> >
> > Dear Reviewer 8UZU,
> >
> > Thanks a lot for your feedback and suggestions. As the discussion period is nearing end, please let us know if our response has adequately addressed your concerns, or if you have any remaining questions. We really appreciate your review!

---

> > > ### Comment · Reviewer_8UZU · 2023-12-03
> > >
> > > Thank you for your responses. Some of my concerns have been addressed. Please incorporate the definition of "control answer" into the manuscript. I will adjust the score.

---

### Official Review · Reviewer_7HFY · 2023-10-31

**Soundness:** 2 fair
**Presentation:** 3 good
**Contribution:** 2 fair
**Rating:** 3
**Confidence:** 3

**Summary:**

To verify whether LLM is able to distinguish truth from falsehood in the training data, this paper proposed a personas hypothesis to analyze the model capability. They conducted plenty of experiments to demonstrate that LLM can make full use of truthfulness data source to generalize truthfulness. Meanwhile, finetuning operation on truthfulness data will improve the model performance and model generalizations to unseen topics and domains. Finally, this paper conducted a detailed analysis about their observations and the limitations of their work.

**Strengths:**

1.	This paper proposed a novel personas hypothesis to verify the LLM capability of distinguish truth and falsehood from contradicting training data, which is very interesting.
2.	This paper conducted various experiments to prove the reasonableness of the proposed hypothesis.
3.	This paper provided a detailed analysis about their work and corresponding limitations.

**Weaknesses:**

1.	The motivation of this paper should be further clarified. The truth and the falsehood in training data of LLMs should be given formal and clear definitions. Is it only those ideas or theories that conform to human intuition or are in the minority that can be called truths? Unclear definitions of truth make this paper not so convincing.
2.	The authors only leveraged additional model to classify the results generated by LLMs to prove whether LLM learns the truth or not. This strategy is not so convincing. More evaluation strategies should be considered, such as causal inference.

**Questions:**

Please refer to the weakness. I am looking forward to the responses from authors. If the responses can convince me, I will change my score.

---

> ### Author Response · Authors · 2023-11-17
> **Response to Reviewer 7HFY**
>
> Thank you for your time to review our paper and to give constructive feedback. We also suggest you go through our general response first since it clarifies our terminology and addresses general issues different reviewers mention.
>
> > The truth and the falsehood in training data of LLMs should be given formal and clear definitions.
>
> Thanks for bringing this up. We have clarified in the general response what we mean by truthful and untruthful text. Specifically, truthful text contains factually correct information —- our focus is on factual questions where there is a widely agreed upon answer (e.g. experts would agree). Untruthful text contains incorrect (i.e. not grounded in evidence) but plausible answers — these are answers which are supported by misinformation in the pretraining data. As an example (from the updated Figure 1), "covid vaccines are ... extremely safe" is truthful text, whereas the text “...millions die from covid injections” is untruthful since it is incorrect but is probably seen in the pretraining data since it is a misconception generated by a source ("Before It's News" in this case).
>
> > The authors only leveraged additional model to classify the results, more evaluation strategies should be considered.
>
> We interpreted this comment in two ways:
>
> (a) ‘additional model’ refers to the probe used to classify true vs false statements; We use probing to predict if the generation will be truthful only from the representation of the question. This means that we can linearly separate truthful and untruthful text generated from different agents — evidence which is consistent with our hypothesis. Probing is an established empirical method to evaluate the presence of representations in the latent space of a model.
>
> (b) ‘additional model’ refers to using GPT-judge as the evaluator of truthfulness. We want to note here that in addition to using GPT-judge (model to evaluate truthfulness), we also do human evaluation for both the datasets (truthfulQA and BigBench-misconceptions) in Sec 2.2. Additionally, we note that there is a strong correlation between both forms of evaluation (model based and human) — both as observed in our work and as reported in previous work (e.g. Lin et al 2021).

---

> > ### Author Response · Authors · 2023-11-22
> >
> > Dear Reviewer 7HFY,
> >
> > Thanks a lot for your feedback and suggestions. As the discussion period is nearing end, please let us know if our response has adequately addressed your concerns, or if you have any remaining questions. We really appreciate your review!

---

### Official Review · Reviewer_CCLc · 2023-11-01

**Soundness:** 3 good
**Presentation:** 4 excellent
**Contribution:** 2 fair
**Rating:** 3
**Confidence:** 3

**Summary:**

This paper proposes a hypothesis to explain why it is possible for LLMs to generate truthful answers given very little training data and how training on truthful examples can help generalize truthful behaviors on unrelated topics.

The hypothesis itself unfortunately is still slightly vague, and one of the main experiments (Section 3) does not represent the reality. If the authors can very clearly explain what their hypothesis is, what a persona is, and discuss how their empirical finding can improve training LLMs to be more truthful, my decision can be changed.

**Strengths:**

This is a very interesting empirical paper that's well organized.
- Important topic -- trying to understand why LLMs are truthful or untruthful has strong societal implications.
- The hypotheses are clearly proposed, and evidence is laid out.
- Well-written and well-organized

**Weaknesses:**

The paper has a few confusing aspects to me.

1. First of all, the terminology "persona" -- my thinking around persona has been mostly influenced by Social Simulacra [1], using constructed persona profiles as prompts for LLMs and controlling what the LLMs generate. At first, I thought the paper was discussing persona in the same way, but from the experiment and even from Figure 1, it is hard to tell what is the "truthful" or "untruthful" persona the paper is referring to. Can the authors explain by persona, do they mean a concrete part of the LLM (i.e., a prompt or an embedding at a particular token position?) To alleviate confusion, maybe choose a slightly different wording?

2. The paper cited Jacob Andrea's LLMs as Agent Models. The hypothesis says, "There exists a group of agents who are more truthful than others," and suggests that "By modeling this truthful persona, language models can." Does this suggest that LLMs model persona and each persona is an agent? The hypothesis does not make enough sense to me. Where is this group of agents? Is it implicitly defined by LLMs' weights? Is it defined by the prompt (hypothesis seems to suggest "Wikipedia and Science" to be personas/agents)? The paper would be hugely improved if authors could clearly define what they mean by agent and by persona.

3. Section 3, the two setups -- the most significant difference is the first setup, two agents are FULLY truthful, and two agents are FULLY dishonest. The other setup is partially truthful. I don't think this setup proves the hypothesis -- A lot of untruthful information, or if we think about conspiracy stories, often have a shroud of truth in the story and then mix them with lies. This truthful persona setup (that supports your hypothesis) is not close to reality -- but please feel free to showcase why you think it's closer to the actual training data of LLMs.

4. Would it be more interesting if the authors could extract these personas and visualize them? Or show how fine-tuning or ICL can actually force LLMs always to adopt a truthful persona. I guess this requires the authors to define what a persona is, but it's very unclear what they mean from this paper.

[1] Park, Joon Sung, et al. "Social simulacra: Creating populated prototypes for social computing systems." Proceedings of the 35th Annual ACM Symposium on User Interface Software and Technology. 2022.

**Questions:**

See weakness section

---

> ### Author Response · Authors · 2023-11-17
> **Response to Reviewer CCLc**
>
> Thank you for your time to review our paper and constructive feedback! We suggest you go through our general response first since it clarifies our terminology and addresses general issues different reviewers mentioned.
>
> > First of all, the terminology "persona" -- my thinking around persona has been mostly influenced by Social Simulacra [1], using constructed persona profiles as prompts for LLMs and controlling what the LLMs generate.
>
> In our general response, we have motivated the choice of the terms in the paper. We chose the terms ‘agent’ to be consistent with the work we build on (Andreas et al), and ‘personas’ as a natural extension of the term.
>
> > from Figure 1, it is hard to tell what is the "truthful" or "untruthful" persona the paper is referring to.
>
> We have updated Figure 1 to reflect what is a "truthful" or "untruthful" persona --- it refers to a latent variable corresponding to a cluster of agents (left part of figure), which the model can use at inference time to generalize truthfully or untruthfully (right side of the figure).
>
> Another way to understand “persona” is that truthful text shares some common features — we didn’t want to call it `features’ because it can imply that these common properties are local (e.g. using specific words), whereas they could be more abstract properties. More generally, our hypothesis is that agents which are truthful (e.g. Wikipedia, Science journal etc.) tend to form a cluster due to common properties like consistency, providing evidence etc. — we refer to this cluster of truthful agents as the truthful persona.
>
> > The paper would be hugely improved if authors could clearly define what they mean by agent and by persona.
>
> We have addressed this in our general response and will clarify in a revised version of the paper.
>
> > LLMs model persona and each persona is an agent?
>
> We elaborate on this in our general response. The short answer is that each persona is not an agent; a persona is a group of agents that share common properties like truthfulness. Both agent and persona are used to explain the data generating process.
>
> > Where are the personas defined in the models?
>
> It is important to highlight that agents and personas are constructs to model the data generating process (i.e. latent variables), and they do not necessarily correspond to physical entities. However, we find evidence that these constructs are useful to explain model behavior. We use our probing results to show that we can decode a “truthful persona” from the latent activations of the model, which suggests the model encodes this information in its hidden states.
>
> > The synthetic setup is not close to reality
>
> We want to clarify a potential misunderstanding in the Sec 3.1 setting. In the first setup, when the two agents (C and D) are fully dishonest, we mean that the probability that the agents are truthful on each operator is small (< 0.2 — sec 3 dataset generation & appendix D.1). Therefore the agents are not always untruthful but they do have a small probability of being truthful. Similarly agents A and B are not fully truthful, but have a small probability of generating an untruthful answer. We think this is a realistic setup, where there is no one source which is 100% truthful or 100% untruthful —- but some sources are much more likely to be truthful and trustworthy (e.g. Wikipedia would have a high probability of being truthful ; Similarly an author who believes in a conspiracy would mostly write untruthful statements to support the conspiracy but that will be mixed in with some amount of truthful information).
>
> > Would it be more interesting if the authors could extract these personas or show how fine-tuning or ICL can actually force LLMs to always adopt a truthful persona
>
> In our experiments (sec 2.1) we extract these personas by training linear probes to distinguish between true and false statements based on hidden states from the model. This is evidence consistent with our hypothesis that LLMs infer persona from context and represent it in the hidden states, which the probe can rely on, i.e. the probe is extracting the persona. However, we do not have interpretable features that represent the persona.
> Also, we show that both finetuning and ICL steer the model to adopt the truthful persona (sec 2.2).

---

> > ### Author Response · Authors · 2023-11-22
> >
> > Dear Reviewer CCLc,
> >
> > Thanks a lot for your feedback and suggestions. As the discussion period is nearing end, please let us know if our response has adequately addressed your concerns, or if you have any remaining questions. We really appreciate your review!

---

### Author Response · Authors · 2023-11-17
**General Response to Reviewers**

We would like to thank all the reviewers for their time and constructive feedback.

In this general response, we want to clarify key concepts that have caused confusion among several reviewers and may have made the contributions less clear.

## Terminology

We first define what we mean by truthful and untruthful text. **Truthful text** is aligned with facts that most domain experts agree. **Untruthful text** is incorrect (i.e. not aligned with facts) but plausible — in particular we consider untruthful text  that is supported by misinformation in the pretraining data.

Our hypothesis relies on two concepts: agents and personas, which we construct to model the generative process of the data. Following Andreas [1], we define **agent** to be the source that generates a particular piece of text.  (e.g. if the text is from the Wikipedia page of Barack Obama then the agent would be Wikipedia). We define **persona** to be a set of agents that share common properties (i.e. they are latent clusters of agents). For example, the following set of agents would form a persona since they share common properties like being consistent, providing evidence, etc. --- {Wikipedia, Science journal, CDC etc.}

Our main hypothesis is that the agents which tend to be truthful form a persona due to common properties like consistency, providing evidence etc. Thus the model can use this persona (cluster of agents) to distinguish true statements from false statements, and generalize truthfulness across different topics.

Potential confusion with “agent” and “persona” from other works: We adopt the definition of “agent” to stay consistent with the work by Andreas we use as a motivation [1]. We are aware that previous work (e.g. [2]) has also used the term “persona” with a definition closer to our definition of “agent”. We tried to look for alternative names, but we believe agent and persona are the ones that best convey our idea. If you think other terms could help clarify our work, we are happy to discuss them over the rebuttal period.

## Contribution and relation with previous work

We clarify here our main contribution.

Past work has:
1. Presented the hypothesis that LLMs model the communicative agent [1]
2. Showed empirical evidence of truthfulness being encoded in the latent space [3,4]

We meant this work to be a bridge to show how the hypothesis from (1) can be extended to explain the observations in (2).

Our persona hypothesis extends the “agent hypothesis” [1] and suggests that LLMs may cluster agents who are truthful (we refer to this cluster as a truthful persona) in the representation space due to common properties. Note that this is non-obvious because it suggests hierarchical modeling of the text generation process (persona -> agent -> text).

If this hypothesis was true, then it would explain results found in empirical observations [3,4]. It is important to highlight that agents and personas are constructs to model the data generating process, and they do not necessarily correspond to physical entities. However, we find evidence that these constructs are useful to explain model behavior.

We run experiments that validate the plausibility of this hypothesis:
We probe the question representation to predict if the generation will be truthful. According to [1] the model could infer the agent from this context, and we show that truthful agents may also be clustered since our probe can distinguish between true and false statements.
We show that finetuning improves truthfulness on unseen topics. This implies that truthful behavior is modeled by the LLM globally (through personas) and not only locally for each fact.

We believe these findings are non-trivial because: (1) they contradict the classic view of LMs that claims that the probability of an answer should be proportional to their frequency in the training data, and not necessarily take into account the agent-based view; and (2) prior work has only focused on methods to elicit truthfulness, but has not tried to explain the mechanism of LLMs modeling truthfulness. It is not obvious why LMs would encode truthfulness since it is not prescribed by the learning objective and there is little naturally-occuring truthfulness label in the training data.

We welcome additional suggestions from the reviewers for more clarity during the discussion period.

## Updated Draft

We have uploaded an updated draft including new results in Appendix B as suggested by one reviewer (oRbY). We plan to revise the draft further to make the definitions of agent and persona more prominent and will upload the final revised version before the discussion period deadline next week.

---

> ### Author Response · Authors · 2023-11-17
> **General Response to Reviewers (continued)**
>
> ### References
>
> [1] Andreas, J. (2022). Language Models as Agent Models. ArXiv, abs/2212.01681.
>
> [2] Park, Joon Sung, et al. "Social simulacra: Creating populated prototypes for social computing systems." Proceedings of the 35th Annual ACM Symposium on User Interface Software and Technology. 2022.
>
> [3] Burns, Collin, et al. "Discovering latent knowledge in language models without supervision." arXiv preprint arXiv:2212.03827(2022).
>
> [4] Li, Kenneth, et al. "Inference-Time Intervention: Eliciting Truthful Answers from a Language Model." arXiv preprint arXiv:2306.03341 (2023).

---

### Author Response · Authors · 2023-11-20
**Updated Draft**

We thank all the reviewers again for their helpful comments and feedback. Based on the suggestions, we have uploaded an updated draft with the following main changes included:
1. More clear, explicit definitions of “agent” and “persona” in introduction (section 1).
2. An updated main figure (figure 1) which more clearly reflects what is an agent, a persona and the relationship between the two. The figure also highlights our main hypothesis of how LLMs model truthfulness via personas.
3. Clarification in the introduction (section 1) of how our work relates to prior work as a bridge between Andreas 2022 [1] and {Burns et al., 2022 [2], Li et al., 2023 [3]}
4. New results as requested by reviewer oRbY in Appendix B — probing along the answer tokens (figure 9) and reporting accuracy in addition to F1 score (figure 8).

[1] Andreas, J. (2022). Language Models as Agent Models. ArXiv, abs/2212.01681.

[2] Burns, Collin, et al. "Discovering latent knowledge in language models without supervision." arXiv preprint arXiv:2212.03827(2022).

[3] Li, Kenneth, et al. "Inference-Time Intervention: Eliciting Truthful Answers from a Language Model." arXiv preprint arXiv:2306.03341 (2023).

---

### Meta-Review · Area_Chair_XExp · 2023-12-05

**Metareview:**

The authors test the hypothesis that personas (different agents producing data in the corpus) can control the truthfulness of a language model, and test this via a probing experiment and a finetuning experiment.

Strengths: the paper poses an interesting abstract hypothesis (personas drive truthfulness) and come up with ways to try to operationalize and test that hypothesis.

Weakness: basically every reviewer agreed that the paper was confusing - not even just at the writing level but more broadly about what exactly is the truthfulness quantity that's being tested, or what personas are more formally.

**Justification For Why Not Higher Score:**

While the authors do valiant work rewriting and reframing the paper, there should probably be a more strutural rethinking of how to present the personas in a clearer, more formal way.

**Justification For Why Not Lower Score:**

N/A

---

### Decision · Program_Chairs · 2024-01-16

Reject